# Technical note:

## An economical apparatus for the observation and harvesting of mineral precipitation experiments with light microscopy

Chris H. Crosby[1], Jake V. Bailey[1]

[1]Department of Earth Sciences, University of Minnesota–Twin Cities, Minneapolis, 55455, USA

*Correspondence to*: Chris H. Crosby (crosb118@umn.edu)

**Abstract.** We describe a small-scale reusable and low-cost double diffusion setup that allows microscopic observation over time for use in mineral precipitation experiments that use organic polymers as a matrix. The setup uniquely accommodates

changes in solution chemistry during the course of an experiment, and facilitates easy harvesting of the precipitates for subsequent analysis.

**Keywords:** Mineral precipitation; 4D imaging; microscopy

## 1 Introduction

Investigations into the influence of organic materials and microbes on authigenic mineral precipitation has transformed our understanding of geosphere/biosphere interactions and improved our understanding of taphonomic process that allow for the preservation of biological remains. The ability to observe nucleation, precipitation and growth over time can provide insight into these processes. However, observation and imaging over the course of an experiment, as well as post-experimental analysis, place strict requirements on the experimental setup, including the following:

1) The setup must provide an approaching flux of counter-ions while simultaneously slowing diffusion sufficiently to avoid the instantaneous precipitation that would inhibit further crystal growth.

2) To enable microscopic observation over time, the setup must fit a microscope stage during the course of an experiment. The diffusion gel within which precipitation proceeds must be transparent to the imaging wavelength, and, for undistorted optical imaging, the region/material of interest should be housed within a planar (not tubular) transparent
housing.

3) For post-experimental analysis of precipitates, the setup must allow harvesting of the materials of interest, which may be both precipitates and various nucleation substrates of interest.

4) Both biological and abiological processes are sensitive to changes in ambient conditions, including temperature, redox conditions and chemistry. The ability to change these conditions expands the usefulness of the apparatus, allowing
exploration of increasingly refined and focused questions.

Here, we describe a reusable, small scale and low-cost Double Diffusion (DD) apparatus that satisfies these requirements and requires only small diffusion gel volumes – a significant advantage when the gel material is expensive or consists of low-volume microbially-produced polymeric substances. The apparatus allows detailed observation of progressive precipitation in situ, an example of which can be seen in Fig. S2 of the Supplementary File.

## 2 Background

A variety of physical setups have been used for decades by chemists investigating mineral precipitation kinetics and are generally one of three types: The Single Diffusion (SD) method, in which an ion-containing gel is overlain with a solution of counter-ions that diffuse into the gel; the Double Diffusion (DD) method, in which solutions of constituent ions are separated by a diffusion gel and into which the ions pass and ultimately meet (Becker et al., 2003); and the Constant Composition (CC) method (Morse, 1974; Tomson and Nancollas, 1978). As the name indicates, the CC method holds the ionic strength of constituent ions constant and allows sensitive observation of the impact of factors other than ionic strength. But for exploring systems relevant to essentially confined environmental systems – such as sediment pores or spaces constrained within polymeric matrices such as those found in sediment or under stromatolitic growth conditions – a diffusion setup is arguably more likely to reflect dynamic in situ conditions, where precipitation leads to falling ion concentrations over time. Hence, diffusion setups are suitable for, and have been used in, studies of biologically mediated precipitation (Becker et al., 2003; Emerson et al., 1994; Hunter et al., 1985).

The DD setup described herein is the result of many iterations and refinements of a setup similar to that described by Kniep et al. (Busch et al., 1999; Kniep and Busch, 1996). It resembles that of Emerson et al., designed to observe the responses of motile microbes to distinct gradients (Emerson et al., 1994), but differs in that this apparatus immobilizes biological material as counter-ions meet across the immobilized biological material.

In this system, the gel functions to both: 1) slow precipitation by retarding ion diffusion rates, and 2) serve as a proxy for microbially-produced polymeric substances, such as EPS (microbial extracellular polymeric substances), a matrix that is ubiquitous in microbial mats and biofilms. Considering the diffusion gel as the primary organic matrix, this setup will also accommodate "secondary organics" such as distinct EPS strands or pellets of microbial culture which can be immobilized by slight heat fixation/adherence to the bottom cover slip of the assembly before addition of the diffusion gel. Staining of secondary organics may also be accommodated. It is small and easily handled and fits unobtrusively in a laboratory refrigerator for low temperature experiments. However extra care will need to be taken when imaging low temperature experiments, as the gel may be heated by the light source. In this case, preliminary tests with the microscope are strongly advised to determine the degree to which this may occur, and methods for ameliorating it (such as blowing cooled air between the light source and apparatus.) Nominal monitoring of gel temperature might be possible by measuring the temperature of the two cover slips by remote sensing, such as may be obtained by an infrared digital laser thermometer.

Experiment goals will dictate protocol details. For instance, after adhering a marine culture, gentle rinsing may be required to remove NaCl precipitates or growth media if their presence would interfere with the goal of the experiment.

A variety of polymeric substances are available for use, including lab standard polymers such as gelatin and agar, or custom organic substrates such as lab-grown extracellular polymeric substances (EPS). The characteristics of each polymer are unique, most significantly in the nature and location of their charge balances – a discussion of which can be found in Kniep et al. (Kniep and Simon, 2007) – and solubility (Whistler, 1973). The reader is referred to the book "Microbial Extracellular Polymeric Substances" (Wingender et al., 1999) for discussion of EPS, and to Silverman & Boskey for a discussion of different polymers and the utility of DD setups in studying biomineralization. They also describe a constant composition DD method comparing different proteins introduced into a gelatin matrix to illustrate their effects on calcium-phosphate biomineralization (Silverman and Boskey, 2004).

## 3 Apparatus

### 3.1 Apparatus description

The active precipitation area of this setup is a thin diffusion gel, ~1 mm thick, sandwiched between a long coverslip and a square coverslip, into which ions are introduced from solution chambers via small channels leading into the gel (Fig. 1). Table 1 lists the required components. The setup block and adaptor/spacer can be easily made in-house. Detailed directions and solution concentrations suitable for reproducing our initial experiments can be found in Supplementary Information. The purchasable components of the setup are readily available.

### 3.2 Apparatus design considerations

The setup described here was designed with a cover slip bottom to allow use in an inverted microscope. Before constructing the setup, take into consideration the microscope that will be used and adjust dimensions as needed. For instance, an upright scope with objective lenses in a rotating turret may require a design with longer channels on either side of the centre bore to avoid contact between the setup and the objective lenses.

If the precipitates are to be harvested from the diffusion gel for additional analysis, the solubility of the diffusion gel material should be taken into account. Gelatin is easily removed by repeated applications of hot water. Other gel materials are likely to require different treatments.

The success of this apparatus requires seals adequate to preclude leakage and evaporation and keep the ion solutions from bypassing the diffusion gel and mixing prematurely. A great deal of trial and error went into this protocol, the results of which are incorporated into the SI step-by-step protocol. Please see SI for details of the sealing method and precautions (§ "Add small cover slip," § "Add setup block," § "Add ion solutions, seal side bores.") Assembly requires practice to achieve a full seal. Experimental details can be found in the SI file.

## 4 Experimental results – an example

Our interest in developing this apparatus and protocol stemmed from our exploration of the influence of organics on the precipitation of calcium phosphates (apatite and its precursor phases.) We designed the apparatus to replicate conditions that might be present under primarily confining conditions such as within the microbial EPS of growing stromatolites or sediment pore spaces.

Early iterations of the apparatus with 3/8-inch gel depths formed a thick opaque cloud of precipitates, precluding optical microscopic imaging of precipitate details. In the assembly being described, the thickness of the gel is determined by the spacer used in forming the adaptor/spacer, as shown in SI Fig. A1, in which a square cover slip is used to create the spacer inset into which the diffusion gel is added. This "thinner" gel thickness allows details of the development and maturation of the precipitation cloud that had been hidden in the deeper gels to be revealed. A similar initially diffuse precipitation cloud forms, but then develops into a much more pronounced band with distinct boundaries before dividing into a number of discrete bands (Liesegang bands) as shown in Fig. 1e. The phenomenon and dynamics of Liesegang banding remains an area of active research (Antal et al., 1999; Stern, 1954; Tripathi et al., 2015) and imaging and analysis of these separating Liesegang bands have shown differences in the size and morphology of the small constituent precipitates.

## 5 Conclusion

The features of this apparatus make it a versatile instrument for experiments in which microscopic observation of the precipitation process is desired. It is small and easily handled and fits unobtrusively in a laboratory refrigerator for low temperature experiments. It requires only small amounts of diffusion gel and can accommodate secondary organics of interest. Experiments can be designed with any desired counter-ion solutions – the solution chemistry, pH and Eh of which can be changed mid-experiment by needle and syringe. When utilized in conjunction with time-lapse microscopy, this apparatus provides an efficient and economical opportunity to observe and document mineral precipitation throughout the process.

**Author contributions:** JVB contributed initial experimental goal. CHC developed the apparatus and protocol. CHC & JVB wrote the paper.

**Competing interests:** The authors declare that they have no conflict of interest.

**Disclaimers:** None

**Acknowledgements**

ESEM imaging and EDS analysis not described herein but utilized in the development of the setup and protocol was performed at the LacCore (National Lacustrine Core Facility), Department of Earth Sciences, University of Minnesota–Twin Cities (UMN). LacCore is funded by NSF and UMN. Portions of this work were funded by NSF grants #3002-1113-

00019448 and #EAR-1057119 to JVB, and a UMN Graduate School Fellowship, Doctoral Dissertation Fellowship, and Department of Earth Sciences Fellowships to CHC. The assistance of Mark Griffith, Elizabeth Ricci, Beverly Chiu, Erica Sheline and Peter Schroedl is gratefully acknowledged.

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

**Table 1**

**Setup components**

| Component | material | dimensions |
|---|---|---|
| Long coverslip | glass | 24 x 60 mm |
| Square coverslip | glass | 18 mm square |
| Setup block | plastic or glass | ~ 26 x ~ 62 mm (thickness: ~3/8 inch or as preferred) |
| Adaptor/spacer | | silicone epoxy ~ 26 x ~ 62 mm (thickness: standard glass slide) |
| Assembly material | sterile bandaging * | > 24 x 60 mm |
| Assembly material | clear tape | ¾-inch |
| Assembly material | Vaseline/lanolin | |

**Table 1: Dimensions of the setup block and adaptor/spacer should be slightly narrower and shorter than the long coverslip.**

**\*Half of a 2-3/8 x 2-3/4-inch 3M Nexcare™ Tegaderm™ bandage, or similar, works well where sterility is desired, but its adhesive surface is designed to allow the escape of moisture. Clear watertight tape will seal it from evaporative loss. The bandage/tape plies can be punctured by needle for exchange of solutions during experimentation and resealed.**

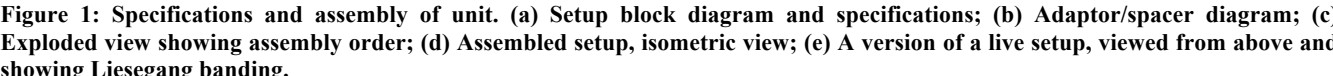

**Figure 1: Specifications and assembly of unit. (a) Setup block diagram and specifications; (b) Adaptor/spacer diagram; (c) Exploded view showing assembly order; (d) Assembled setup, isometric view; (e) A version of a live setup, viewed from above and showing Liesegang banding.**

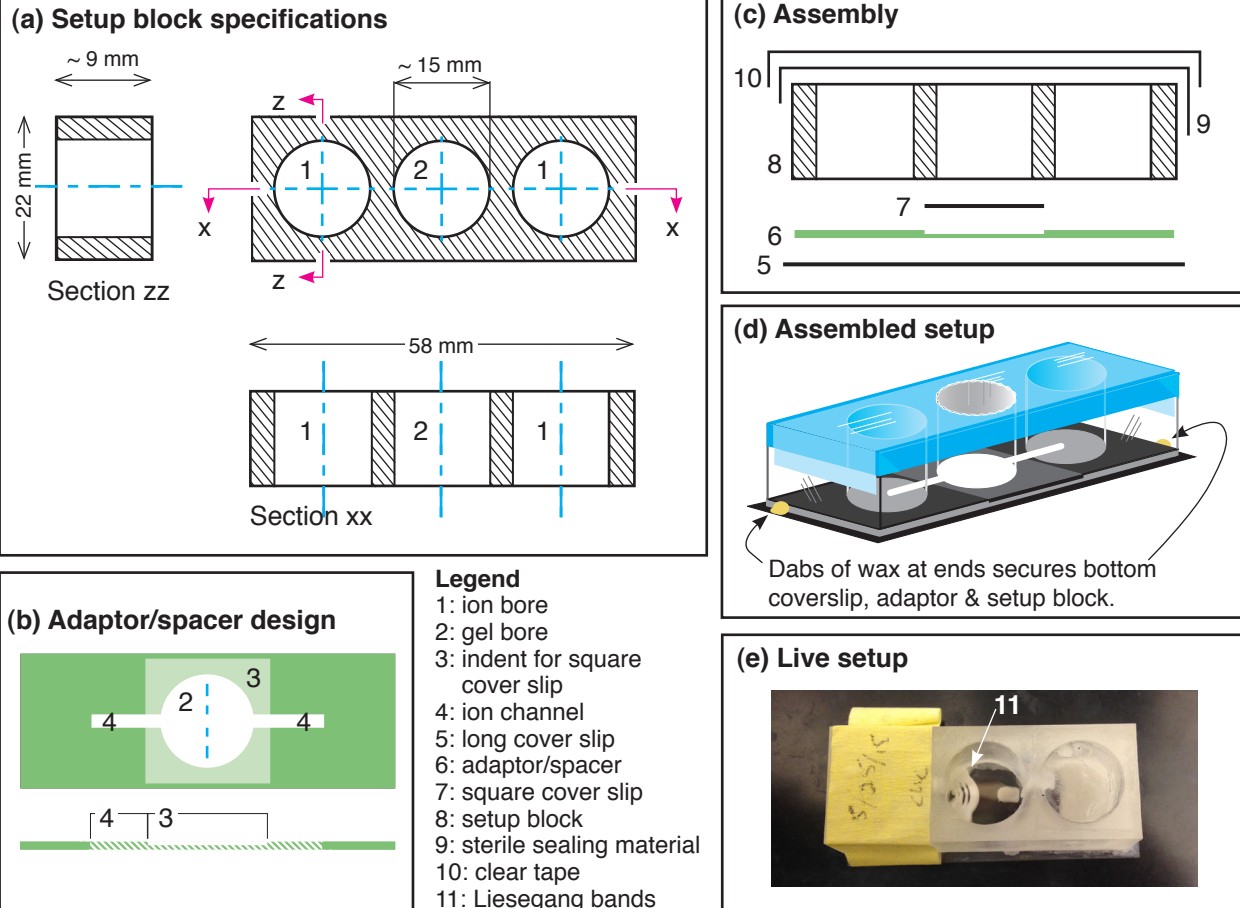

**(a) Setup block specifications**

~ 9 mm
22 mm
Section zz

~ 15 mm
z
x — 1 — 2 — 1 — x
z

58 mm
Section xx
1 — 2 — 1

**(b) Adaptor/spacer design**

4 — 2 — 4

4 — 3

**(c) Assembly**

10
8 — 9
7
6

**(d) Assembled setup**

Dabs of wax at ends secures bottom
coverslip, adaptor & setup block.

**(e) Live setup**

11

**Legend**

1: ion bore
2: gel bore
3: indent for square
   cover slip
4: ion channel
5: long cover slip
6: adaptor/spacer
7: square cover slip
8: setup block
9: sterile sealing material
10: clear tape
11: Liesegang bands