# Peer review of "Technical note"

_Biogeosciences, 2016_

## Referee Comment (RC1) · Anonymous Referee #1 · 25 Jan 2017

The manuscript from C. Crosby and J. Bailey entitled "An economical apparatus for the observation and harvesting of mineral precipitation experiments with light microscopy" provides a brief introduction to a novel apparatus designed to enable time-resolved imaging of mineral precipitate formation. The supplement to the manuscript provides brief instructions on the construction of the apparatus, and should be sufficient for replication of the design. While the manuscript discusses the utility of this apparatus for microscopic imaging, it unfortunately lacks any photomicrographs of mineral formation, making the objective assessment of the conclusions challenging, even in the scope of this brief technical note.

[Figure]

The figures provided are useful, but of very low resolution as supplied for review. Higher resolution is necessary, particularly for figure 1E.

While the manuscript is well written and supported by its cited references, I do not believe it can be accepted without the demonstration of functionality, perhaps by the inclusion of photomicrographs in the supplemental information.

---

## Referee Comment (RC2) · Anonymous Referee #2 · 29 Jan 2017

This is a useful effort and it would be valuable contribution to the literature. However, some details and other additions would improve its impact for the reader.

Page 2 A reference to add to or replace the Tomson and Nancollas reference is: Morse, J. W. "Dissolution kinetics of calcium carbonate in sea water; III, a new method for the study of carbonate reaction kinetics." American Journal of Science 274.2 (1974): 97-107. Morse was the first to propose this method.

Another reference to add about precipitating a mineral within an extracellular matrix by diffusion is Hunter, G. K., et al. "Inhibition of hydroxyapatite formation in collagen gels

by chondroitin sulphate." Biochemical Journal 228.2 (1985): 463-469.

For the paragraph that begins on line 20, it would be useful to break this down into at least two sentences.

Line 25, consider replacing "flow" with "diffusion"

Page 3 Line 5: It might also be worth pointing out in some more detail in your text that the Silverman article reviews many diffusion studies within different extracellular matrices with the goal of studying biomineralization.

Line 12: Some more experimental details about the setup, for example some solution concentrations and diffusion times with your setup, would be very helpful for the reader.

Line 19: The example given for dissolving gelatine is quite aggressive; contact with hot water may alter the minerals and/or their precursors generated by the experiment. The statement that other gel materials will require different treatments is quite general; perhaps a statement suggesting that extracellular matrix removal methods must also consider the stability of the precipitation products would be more helpful.

Line 21: The discussion about sealing strategies would benefit from more details. For example, describing a strategy with which you found success would be useful.

Line 29: Please define "thinner" for the reader, or provide example values.

Page 4 Lines 1-5: Providing the details of an experiment, such as how the ECM was set up and inserted into the cell, what its solution composition was, and what the solution compositions were for the other solutions would help the reader to begin testing this method with a system that should provide initially positive results.

Line 11: Please explain how the needles and syringes are used in more detail. Must the user remove the source solutions and then replace them with the next solutions? Can the use of a needle and syringe change the fluid dynamics appreciably for your system?

Figure 1d) The assembly detail could be more clear. The different stages of assembly, to the point of adding the ECM and solutions of interest, would be helpful for the reader to repeat your method.

Figure 1e) The scale of this image is not adequate for observing the Liesegang banding easily. Consider including an inset with a higher magnification.

Question: Do thinner gels heat more rapidly from the light source? Is there a method for monitoring this temperature change, and/or variation within the cell?

---

## Author Comment (AC2) · 20 Feb 2017

10   **Anonymous Referee ##2**

This is a useful effort and it would be valuable contribution to the literature. However, some details and other additions would improve its impact for the reader.

15

Page 2 A reference to add to or replace the Tomson and Nancollas reference is: Morse, J. W. "Dissolution kinetics of calcium carbonate in sea water; III, a new method for the study of carbonate reaction kinetics." American Journal of Science 274.2 (1974): 97-107. Morse was the first to propose this method.

   We will add this important reference to the final draft, as well as the one suggested below:

20   Another reference to add about precipitating a mineral within an extracellular matrix by diffusion is Hunter, G. K., et al. "Inhibition of hydroxyapatite formation in collagen gels by chondroitin sulphate." Biochemical Journal 228.2 (1985): 463-469.

For the paragraph that begins on line 20, it would be useful to break this down into at least two sentences.

25   Agreed – will do so in final draft.

Line 25, consider replacing "flow" with "diffusion"

   Yes – good catch. Thanks.

Page 3 Line 5: It might also be worth pointing out in some more detail in your text that the Silverman article reviews many diffusion studies within different extracellular matrices with the goal of studying biomineralization.

30   Will add: "Silverman and Boskey utilized the DD method to compare a variety of extracellular matrices in HAP biomineralization, as relevant to our understanding of physiologic  HAP precipitation affected by a variety of proteins (Silverman & Boskey, 2004.)

Line 12: Some more experimental details about the setup, for example some solution concentrations and diffusion times with your setup, would be very helpful for the reader.

35   Will refer reader to SI for additional info on solution concentrations and diffusion times.

To be added to SI:
Apparatus diffusion gel material and solution concentrations can be altered as required for different experiments, but the conditions under which this apparatus was developed, and those used to produce figure 1E, are as follows:
• gelatin: type A, 1 g/10mL water, pH ~4
• cation solution: 0.133 $\underline{M}$ $Ca^{2+}$: $CaCl_2 \bullet 2H_2O$, pH ~8
• anion solution: 0.08 $\underline{M}$ $(PO_4)^{3-}$, $NaHPO_4$, pH ~8
• anion solution: 0.027 $\underline{M}$ $F^-$, $KF \bullet 2H_2O$, pH ~8

Nascent precipitation observed within ~30 hours.

Line 19: The example given for dissolving gelatine is quite aggressive; contact with hot water may alter the minerals and/or their precursors generated by the experiment. The statement that other gel materials will require different treatments is quite general; perhaps a statement suggesting that extracellular matrix removal methods must also consider the stability of the precipitation products would be more helpful.

The primary intent of this manuscript is to describe a protocol and apparatus for microscopic observation of mineral precipitation over time. It is intended that experimental details will represent conditions as desired by the experimenter. The conditions under which this apparatus was developed are given in the opening note of the SI. In particular, gelatin was chosen as a diffusion gel because its gel is transparent and amenable to optical observation and imaging, and because it is soluble in hot water. For work in which a different diffusion material is to be used, if extraction of precipitates is desired it will be important to consider both the solubility requirements of the diffusion material and the stability of precipitates and precursor molecules under whichever extraction method is required.

Line 21: The discussion about sealing strategies would benefit from more details. For example, describing a strategy with which you found success would be useful.

Will add: "A great deal of trial and error went into this protocol, the results of which are incorporated into the SI step-by-step protocol. Please see SI for details of sealing method and precautions (§ 'Add small cover slip,' 'Add setup block,' 'Add ion solutions, seal side bores.')"

Line 29: Please define "thinner" for the reader, or provide example values.

Will replace beginning of sentence with "In the assembly being described, the thickness of the gel is determined by the spacer used in forming the adaptor, as shown in SI Figure A1, in which a square cover slip is used to create the adaptor inset into which the diffusion gel will be added."

Page 4 Lines 1-5: Providing the details of an experiment, such as how the ECM was set up and inserted into the cell, what its solution composition was, and what the solution compositions were for the other solutions would help the reader to begin testing this method with a system that should provide initially positive results.

Insertion of the diffusion gel is described in SI (§ 'Mix gel' and 'add gel.')
Solution details will be added to SI, as noted in above comment.

Line 11: Please explain how the needles and syringes are used in more detail. Must the user remove the source solutions and then replace them with the next solutions? Can the use of a needle and syringe change the fluid dynamics appreciably for your system?

Will add descriptive text: "Addition to and/or removal and replacement initial solution can be done under sterile conditions in a biological hood as follows: Using a sterilized blade, carefully cut away the sealing material over one of the solution wells. Use a sterile needle and syringe to remove the original solution and a different sterile needle and syringe to introduce a new solution. Reseal with Tegaderm™ or similar sterile material, and seal against evaporation with clear watertight tape as described in SI § 'Add ion solutions, seal side bores'.

5

Figure 1d) The assembly detail could be more clear. The different stages of assembly, to the point of adding the ECM and solutions of interest, would be helpful for the reader to repeat your method.

We feel that this is well described in the assembly steps of the SI. Please indicate steps which would benefit by added description.

10    Figure 1e) The scale of this image is not adequate for observing the Liesegang banding easily. Consider including an inset with a higher magnification.

Final manuscript will include a higher resolution image (see image, below.)

[Figure]

15    Question: Do thinner gels heat more rapidly from the light source? Is there a method for monitoring this temperature change, and/or variation within the cell?

Will add (page 4, ~line 9) "Extra care will need to be taken when imaging the apparatus for low temperature experiments, as the gel may be heated by the light source. In this case, preliminary tests with the microscope are strongly advised to determine the degree to which this may occur, and methods for ameliorating it (such as blowing

20    cooled air between the light source and apparatus.) Nominal monitoring of gel temperature might be possible by measuring the temperature of the two coverslips by remote sensing. Such as may be obtained by an infrared digital laser thermometer.

25

---

## Author Response (AR1)

The manuscript from C. Crosby and J. Bailey entitled "An economical apparatus for the observation and harvesting of mineral precipitation experiments with light microscopy" provides a brief introduction to a novel apparatus designed to enable time-resolved imaging of mineral precipitate formation. The supplement to the manuscript provides brief instructions on the construction of the apparatus, and should be sufficient for replication of the design. While the manuscript discusses the utility of this apparatus for microscopic imaging, it unfortunately lacks any photomicrographs of mineral formation, making the objective assessment of the conclusions challenging, even in the scope of this brief technical note. The figures provided are useful, but of very low resolution as supplied for review. Higher resolution is necessary, particularly for figure 1E.

The figures will be supplied in higher resolution for final draft (see response to Reviewer #02 for Figure 1E at higher resolution.) [replaced low-rez version with a high-rez version in the newly named Figure 1 – revised]

While the manuscript is well written and supported by its cited references, I do not believe it can be accepted without the demonstration of functionality, perhaps by the inclusion of photomicrographs in the supplemental information.

The following images and caption will be added to the SI. [Added at end of SI as Figure S2.]

[Figure]

Figure A2: Photomicrographs of a complex object that precipitated in the diffusion gel of the described apparatus. Image taken on an Olympus IX inverted microscope with DP73 camera. Left image is overview of the precipitated object (scale bar = 100 um.) Internal details shown at right (scale bar = 20 um.)

Biogeosciences Discuss.,

doi:10.5194/bg-2016-488-RC2, 2017

**Authors' response to reviewer #02 comments**

**Re: Interactive comment on "Technical note: An economical apparatus for the observation and harvesting of mineral precipitation experiments with light microscopy" by Chris H. Crosby and Jake V. Bailey**

10    **Anonymous Referee ##2**

This is a useful effort and it would be valuable contribution to the literature. However, some details and other additions would improve its impact for the reader.

Page 2 A reference to add to or replace the Tomson and Nancollas reference is: Morse, J. W. "Dissolution kinetics of calcium carbonate in sea water; III, a new method for the study of carbonate reaction kinetics." American Journal of Science 274.2 (1974): 97-107. Morse was the first to propose this method.

    We will add this important reference to the final draft, as well as the one suggested below:
20    [Added at (new) p2, line 10]

Another reference to add about precipitating a mineral within an extracellular matrix by diffusion is Hunter, G. K., et al. "Inhibition of hydroxyapatite formation in collagen gels by chondroitin sulphate." Biochemical Journal 228.2 (1985): 463-469.

    [Added - see p2, line 16]

25

For the paragraph that begins on line 20, it would be useful to break this down into at least two sentences.

    Agreed – will do so in final draft.
    [Reworded - see p2, lines 17-20]

Line 25, consider replacing "flow" with "diffusion"

30    Yes – good catch. Thanks.
    [Done – see p2, line 22]

Page 3 Line 5: It might also be worth pointing out in some more detail in your text that the Silverman article reviews many diffusion studies within different extracellular matrices with the goal of studying biomineralization.

    Will add: "Silverman and Boskey utilized the DD method to compare a variety of extracellular matrices in HAP
35    biomineralization, as relevant to our understanding of physiologic  HAP precipitation affected by a variety of proteins
    (Silverman & Boskey, 2004.)

[Added language to incorporate this information – see p3, lines 7-10]

Line 12: Some more experimental details about the setup, for example some solution concentrations and diffusion times with your setup, would be very helpful for the reader.

Will refer reader to SI for additional info on solution concentrations and diffusion times.
[Added – see p3, lines 15, 16 and p3, line 29]

To be added to SI:
Apparatus diffusion gel material and solution concentrations can be altered as required for different experiments, but the conditions under which this apparatus was developed, and those used to produce figure 1E, are as follows:
• gelatin: type A, 1 g/10mL water, pH ~4
• cation solution: 0.133 $\underline{M}$ $Ca^{2+}$: $CaCl_2 \cdot 2H_2O$, pH ~8
• anion solution: 0.08 $\underline{M}$ $(PO_4)^{3-}$, $NaHPO_4$, pH ~8
• anion solution: 0.027 $\underline{M}$ $F^-$, $KF \cdot 2H_2O$, pH ~8

Nascent precipitation observed within ~30 hours.
[This information was added at the beginning of the SI, lines 5 - 11]

Line 19: The example given for dissolving gelatine is quite aggressive; contact with hot water may alter the minerals and/or their precursors generated by the experiment. The statement that other gel materials will require different treatments is quite general; perhaps a statement suggesting that extracellular matrix removal methods must also consider the stability of the precipitation products would be more helpful.

The primary intent of this manuscript is to describe a protocol and apparatus for microscopic observation of mineral precipitation over time. It is intended that experimental details will represent conditions as desired by the experimenter. The conditions under which this apparatus was developed are given in the opening note of the SI. In particular, gelatin was chosen as a diffusion gel because its gel is transparent and amenable to optical observation and imaging, and because it is soluble in hot water. For work in which a different diffusion material is to be used, if extraction of precipitates is desired it will be important to consider both the solubility requirements of the diffusion material and the stability of precipitates and precursor molecules under whichever extraction method is required.
[The 'notes' section at the start of Supplement C has been revised to include this information]

Line 21: The discussion about sealing strategies would benefit from more details. For example, describing a strategy with which you found success would be useful.

Will add: "A great deal of trial and error went into this protocol, the results of which are incorporated into the SI step-by-step protocol. Please see SI for details of sealing method and precautions (§ 'Add small cover slip,' 'Add setup block,' 'Add ion solutions, seal side bores.')"
[Added – see p3, lines 27-29]

Line 29: Please define "thinner" for the reader, or provide example values.

Will replace beginning of sentence with "In the assembly being described, the thickness of the gel is determined by the spacer used in forming the adaptor, as shown in SI Figure A1, in which a square cover slip is used to create the adaptor inset into which the diffusion gel will be added."
[The language at this point was changed to incorporate the above – see p4, lines 9-10]

Page 4 Lines 1-5: Providing the details of an experiment, such as how the ECM was set up and inserted into the cell, what its solution composition was, and what the solution compositions were for the other solutions would help the reader to begin

testing this method with a system that should provide initially positive results.

Insertion of the diffusion gel is described in SI (§ 'Mix gel' and 'add gel.')
[No changes made to existing text]
Solution details will be added to SI, as noted in above comment.
5      [Added to beginning of SI]

Line 11: Please explain how the needles and syringes are used in more detail. Must the user remove the source solutions

and then replace them with the next solutions? Can the use of a needle and syringe change the fluid dynamics appreciably

for your system?

Will add descriptive text: "Addition to and/or removal and replacement initial solution can be done under sterile
10     conditions in a biological hood as follows: Using a sterilized blade, carefully cut away the sealing material over one of
the solution wells. Use a sterile needle and syringe to remove the original solution and a different sterile needle and
syringe to introduce a new solution. Reseal with Tegaderm™ or similar sterile material, and seal against evaporation
with clear watertight tape as described in SI § 'Add ion solutions, seal side bores'.
[A new section 'Changing ion solutions mid-experiment' was added to the SI – see SI p5, lines 8-13]

15   Figure 1d) The assembly detail could be more clear. The different stages of assembly, to the point of adding the ECM and

solutions of interest, would be helpful for the reader to repeat your method.

We feel that this is well described in the assembly steps of the SI. Please indicate steps which would benefit by added
description.
[We made no changes]

20   Figure 1e) The scale of this image is not adequate for observing the Liesegang banding easily. Consider including an inset

with a higher magnification.

Final manuscript will include a higher resolution image (see image, below.)
[Replaced low-rez version with a high-rez version in the newly named "Figure 1 – revised"]

[Figure]

25

Question: Do thinner gels heat more rapidly from the light source? Is there a method for monitoring this temperature change, and/or variation within the cell?

[revised manuscript text omitted]

Diffusion gel material and solution concentrations can be altered as required for different experiments, but the conditions under which this apparatus was developed, and those used to produce figure 1E, are as follows:

- gelatin: type A, 1 g/10mL water, pH ~4
- cation solution: 0.133 M $Ca^{2+}$: $CaCl_2 \cdot 2H_2O$, pH ~8
- anion solution: 0.08 M $(PO_4)^{3-}$, $NaHPO_4$, pH ~8
- anion solution: 0.027 M $F^-$, $KF \cdot 2H_2O$, pH ~8

Nascent precipitation is observed within ~30 hours.

**Supplement A: Setup block & adaptor construction**

Note: The width and length of the setup block and adaptor/spacer should be slightly smaller than the long coverslip. The dimensions shown in Fig. 1 assume a standard long coverslip size of 24 x 60 mm. The bore diameter should be ~3–4 mm less than a side of the square coverslip. The bore size shown in Fig. 1 assumes an 18 x 18 mm square coverslip. Revise the dimensions as required to accommodate differently sized cover slips.

Setup block
Machine the **setup block** as indicated in Fig. 1a.

Adaptor/spacer
**material needed**
- two clear, rigid flat surfaces ("sheets") such as 8 x 10 inch acrylic or glass sheets
- two microscope slides (of same thickness)
- one square coverslip
- silicone epoxy molding material (Castin'Craft® EasyMold silicone putty, or similar)
- cutting blade, such as an X-acto® blade or similar
- hole punch, 5/8-inch (Recollections™, or similar)

**Molding procedure**
- Position a square coverslip and two slides on one of the sheets as shown in Fig. S1a. The slides will serve as spacers to set the thickness of the adaptor/spacer and their positioning isn't critical, but they should be at least ~ 40 mm apart. The square coverslip will form an indent in the adaptor to accept a square coverslip in the assembled unit.
- Mix a small volume of silicone epoxy, per instructions.
- Form the epoxy into a cylinder and gently press it over the coverslip between the slides, avoiding air bubbles under the epoxy.
- Place the second sheet on the epoxy and press the epoxy into a sheet the thickness of the slides, again observing to avoid air bubbles between the sheets and the epoxy.
- Allow epoxy to set, per instructions.
- Remove the top sheet, pick up the molded epoxy and carefully remove the square coverslip.
- Put the epoxy, indentation side up, back on the sheet and cut out a slot, as shown in Fig. S1b(a)
- Insert a piece of paper into the hole punch as far as it will go without bending, and punch a hole in it Fig. S1b(b).
- Mark off the distance from the center of the hole to the feed-edge of the paper.

❏ Mark off the distance from the center of the hole to the feed-edge of the paper.

❏ Trim one length of the adaptor/spacer so that when it has been fully inserted into the punch a hole will be punched directly in the center of the indent as shown in Fig. S1b(c).

❏ Insert the adaptor/spacer into the hole punch, aligned side-to-side so that the hole will be punched in the center of the indentation Fig. S1b(d). (Practice positioning a sheet of paper in the hole punch before punching the adaptor.)

❏ Trim the adaptor to the size of the setup block Fig. S1b(e).

❏ Check the position of the hole and slot by holding it against a setup block.

**Supplement B: Preassembly preparation**

**Addition of secondary organics**

❏ Secondary organic material (an organic substrate other than the gel material) can be immobilized before assembly of the setup. Place the material in the center of the long coverslip and immobilize/adhere it by allowing it to air dry or quickly passing it over a flame. Rinse as needed to remove media precipitates. Staining of organics may also be accommodated.

**Focusing aids (optional)**

❏ Holding a square cover slip securely against a flat surface, carefully scribe a small 'L' extending from the middle of one side. Rinse off glass scrapings. The combined position and direction of the L will indicate which side is up – the scribed surface will be positioned against the gel and can be used to locate the 'top' surface of the gel layer under the microscope.

❏ Scribe another 'L' in the long cover slip parallel to the length, ~5-6 mm from the edge to indicate the 'bottom' outer surface of the gel layer. Rinse off glass scrapings. In the finished setup, these two scribe marks should overlap near the outer area of the gel, with the scribed surfaces both in contact with the gel surface.

Chris Crosby 3/17/2017 3/17/17 8:25 PM

Chris Crosby 3/17/2017 3/17/17 8:26 PM

Chris Crosby 3/17/2017 3/17/17 8:26 PM

**Supplement C: Assembling and activating the apparatus**

Note: The steps listed here assume the use of gelatine as the primary organic polymer (the diffusion gel.) In developing this apparatus and protocol, gelatin was chosen as a diffusion gel because its gel is transparent and amenable to optical observation and imaging, and because it is soluble in hot water. It is intended that experimental details will represent conditions as desired by the experimenter, such as use of a different diffusion polymer. For work in which a different diffusion material is to be used, if extraction of precipitates is desired it will be important to consider both the solubility requirements of the diffusion material and the stability of precipitates and precursor molecules under whichever extraction method is required.

Ion solutions can be changed mid-experiment to alter solution chemistry, pH, Eh, etc. as desired. Addition to and/or removal and replacement of the initial ion solution(s) can be done under sterile conditions in a biological hood (see § "Changing ion solutions mid-experiment," below.)

Some of these assembly steps are labelled "skill steps." Best results are attained with practice before attempting to assemble and activate a live setup. Components are joined by a layer of Vaseline™. If Vaseline doesn't stand up to your handling, thicken it by adding small amounts of paraffin and lanolin and melting them together. Paraffin will cause the mixture to set more solidly, but too much paraffin may cause it to set so quickly that you will have to work fast, and may make the material brittle and more prone to fracture during experiment handling. Alternatively, a dab of candle wax applied to each end of the assembly (Fig. 1d) can immobilize the setup block–adaptor–long cover slip assembly.

**Materials required for assembly** (assumes gelatin as primary polymer)
- ❏ setup components
- ❏ suction tool, such as a Model Pal™ Suction Handling Tool, or similar
- ❏ hot plate with magnetic stir capacity
- ❏ small stir bar
- ❏ 70% ethanol in a small container for sterilizing various parts and tools
- ❏ small beaker for mixing gelatin
- ❏ gelatin powder
- ❏ ≥ 10 mL sterile water, for gelatin
- ❏ Vaseline®
- ❏ 2 of: 2 x 3-inch glass slide, or similar
- ❏ 3 of: 3 mL syringes with needles
- ❏ disposable pipette
- ❏ pointed tweezers
- ❏ sharp small blade
- ❏ fine metal probe
- ❏ cation solution, pH adjusted as needed
- ❏ anion solution, pH adjusted as needed
- ❏ small candle
- ❏ ~10 mL sterile water, for testing watertightness

**Initial steps: mix gel**  [Assumes gelatin as diffusion gel. Revise as needed for a different gel.]

❑ Before turning the hot plate on, clean its surface, sterilize it with 70% ethanol and allow it to dry.
❑ Place the tweezers tip, probe tip, coverslips, two 2 x 3-inch slides, an adaptor/spacer & setup block in a small container of 70% ethanol to sterilize them.
5 ❑ Turn hot plate to ~30C.
❑ In a small beaker, heat nanopure water for gel.
❑ Measure out gel powder, add to heated nanopure water, add small stir bar and mix.
❑ Dip the suction tool tip into the ethanol to sterilize it and use it to transfer the square coverslip, adaptor/spacer & setup block to the hot plate. Move the 2 x 3-inch slides to the bench top, and place the long coverslip on one of them.
10 ❑ Wipe a thin even layer of Vaseline® onto bottom (flat) surface of the adaptor/spacer.
❑ Center the adaptor onto the long coverslip and press them together.
❑ Verify the seal by the appearance of the adapter material against the long coverslip. Air bubbles should look lighter or darker than well sealed areas and should be avoided where they could allow leakage.
❑ Check the mixing gel and slow the stir rod to release any air bubbles entrained in it.
15 ❑ Remove the tweezers from the ethanol to dry.

**Add gel [skill step]**

❑ Apply Vaseline along the edges of the recessed portion of the adaptor/spacer, avoiding the channels and the gel.
❑ [skill step] Pull ~1 mL of gel into one of the syringes, minimizing entrained air bubbles (pull the syringe plunger slowly,
20 release the plunger and allow it to stop moving before removing the needle tip from the gel.)
❑ [skill step] Extrude a small volume of gel into the center bore to a slight convex meniscus.
❑ Immediately remove bubbles from the gel as needed, using the tweezers in a horizontal 'cutting' motion to pick up and remove bubbles.

25 **Add small cover slip**

❑ Check that the Vaseline along the edges of the recessed portion hasn't been rubbed off.
❑ [skill step] Use the suction tool to pick up the square coverslip and press it over the gel and into the recessed portion of the adaptor/spacer. The gel should fill the volume beneath the square coverslip and a small volume of gel should push out the adaptor/spacer channels.
30 ❑ Check that the square coverslip is sealed against the Vaseline® on the adaptor.
❑ [skill step] Verify that the channels are both clear of Vaseline®. If needed, use a fine probe to clear them of any Vaseline®. Gel in the channels is OK.

**Add setup block**

35 ❑ Apply a layer of Vaseline® onto the bottom of the setup block.
❑ [skill step] Place the setup block on the assembly. Apply downward pressure (only) as needed to get a complete seal (sideward pressure will cause the components to slip out of alignment.) See that seal is complete and neither cover slip has broken.
❑ To stabilize the long coverslip against the setup block when it is handled, light a candle and place a small drop of wax on
40 the setup block side of each end of the long coverslip where it meets the setup block (Fig. 1d).
❑ Gently remove any extraneous Vaseline® from the exterior and place the unit on a clean surface.

**Add ion solutions, seal side bores**

❑ Pipette sterile water into one of the side bores, wait and watch that no water leaks out of the unit or into the other bore. If
45 there is leakage, repair the seal or start over. Once confident that there is no leakage pour the water out and dry the unit.
❑ Repeat for other side.
❑ Add ion solutions to the side bores and label them.
❑ Wick off any water on the setup block exterior.
❑ Cut a section of sterile bandage material, leaving the backing paper on. If using 2-3/8 x 2-3/4-inch 3M Nexcare™
50 Tegaderm™, cut it lengthwise and set one of the halves aside.

❑ Remove the backing, stretch the bandage across the top of the three bores and press it onto the setup block to seal.
❑ Scotch Tape™ (or similar) across the length of the bandage to preclude evaporation through the bandage material.
❑ Use a small sharp blade to cut the tape+Tegaderm™ away from the center bore.
❑ Recheck all seals.
5 ❑ The setup is now activated. Label the setup and place it on a paper towel or similar to detect leakages.
❑ Image as desired, handling carefully so as not to break the Vaseline® seals.

**Supplement D: Harvesting procedure**

10 Note: Once the gel is exposed to air it will begin to harden, and chipping gel off coverslips may break them into slivers. The

gel can be kept soft in cool water. When cutting portions of gel away from the coverslip, place the coverslip against a firm

flat surface.

**Materials needed**
❑ thin X-acto™ blade or similar
15 ❑ dissecting microscope
❑ container for waste ion solutions
❑ small container of cool DI water
❑ small squeeze bottle of DI water
❑ beaker of hot DI water
20 ❑ small (~5–25 µL) pipet and tips
❑ prepared SEM stubs or similar for follow-up analysis and/or imaging

**Procedure**
❑ Open/remove the tape over the ion solutions and pour them into a disposal container.
25 ❑ Working over a catch tray, separate the long coverslip from the adaptor/spacer by gently sliding a thin X-acto™ blade between them and applying a slight twisting pressure against the long coverslip. It should bend a bit and allow the seal to break. Hold the unit so it reflects overhead light to see where the seal is broken – it will be lighter than the intact seal.
❑ Using the same procedure, remove the adaptor/spacer from the setup block. Only the square coverslip and gel plug should remain attached to adaptor/spacer. Leaving the gel plug attached to the cover slip will make it easier to handle. Place them
30 on a firm even surface, such as the stage of a dissecting microscope.
❑ Cut away portions of interest and place on two-sided carbon tape on an SEM stub or similar.
❑ Rinse gel off the precipitates by repeatedly pipetting small amounts of hot water onto the harvested precipitates, letting the gel melt, and then removing the melted gel + water. Repeat as needed to assure removal of gel from precipitates.
❑ For gels with smaller or more numerous precipitates: Placed the gel plug in an epitube of hot water, immerse it in a hot
35 water bath, shake it gently and hold it vertically in the hot water bath to allow precipitates to settle. Slowly remove supernatant by pipette. Repeat as needed to assure removal of gel from precipitates. Remove the final, rinsed precipitates + bottom rinse water from the epitube by pipette and place on a carbon-taped stub. Let the water evaporate and further rinse the precipitates on the stub as described above.

40

[Figure]

[Figure]

**(a) Adaptor moulding layout**

1   1

**Legend**
1: microscope slides
2: square cover slip
3: adaptor epoxy
dashed box: adaptor outline

**(b) Cutting sequence for finishing the adaptor**

**[a]**
cut slit ~2 mm
wide down center
of indent, extend
slit ends past the
cover slip indent

**[b]**
insert a piece of paper into
the hole punch as far as it
will go without bending,
punch a hole in
it and measure
the distance from
center of hole to
the feed-edge of
the paper

**[c]**
trim one length of
the adaptor so a hole
will be punched in the
center of the indent
with the adaptor fully
inserted into the punch

**[d]**
insert adapter into
hole puncher and
punch hole

**[e]**
trim to
size of
setup
block

**Figure S1: Forming the adaptor/spacer.** **(a) Adaptor moulding layout; (b) Cutting sequence for finishing the adaptor.**

Chris Crosby 3/17/2017 3/17/17 8:44 PM

[Figure]

**Figure S2: Photomicrographs of a complex object that precipitated in the diffusion gel of the described apparatus. Image taken on an Olympus IX inverted microscope with DP73 camera. Left image is overview of the precipitated object (scale bar = 100 um.) Internal details shown below (scale bar = 20 um.)**

[Figure]